# Caveolin-1-Related Intervention for Fibrotic Lung Diseases

**DOI:** 10.3390/cells12040554

**Published:** 2023-02-09

**Authors:** Sreerama Shetty, Steven Idell

**Affiliations:** Texas Lung Injury Institute, Department of Cellular and Molecular Biology, The University of Texas Health Science Center at Tyler, 11937 US Highway 271, Tyler, TX 75708, USA

**Keywords:** Caveolin-1, p53, IPF, AECs, fLfs, CSP7, pulmonary fibrosis, dry powder inhalational therapy, interstitial lung diseases

## Abstract

Idiopathic pulmonary fibrosis (IPF) is a progressive and fatal interstitial lung disease (ILD) for which there are no effective treatments. Lung transplantation is the only viable option for patients with end-stage PF but is only available to a minority of patients. Lung lesions in ILDs, including IPF, are characterized by alveolar epithelial cell (AEC) senescence and apoptosis and accumulation of activated myofibroblasts and/or fibrotic lung (fL) fibroblasts (fLfs). These composite populations of fLfs show a high rate of basal proliferation, resist apoptosis and senescence, and have increased migration and invasiveness. They also more readily deposit ECM proteins. These features eventuate in progressive destruction of alveolar architecture and loss of lung function in patients with PF. The identification of new, safer, and more effective therapy is therefore mandatory for patients with IPF or related ILDs. We found that increased caveolin-1 and tumor suppressor protein, p53 expression, and apoptosis in AECs occur prior to and then with the proliferation of fLfs in fibrotic lungs. AECs with elevated p53 typically undergo apoptosis. fLfs alternatively demonstrate strikingly low basal levels of caveolin-1 and p53, while mouse double minute 2 homolog (mdm2) levels and mdm2-mediated degradation of p53 protein are markedly increased. The disparities in the expression of p53 in injured AECs and fLfs appear to be due to increased basal expression of caveolin-1 in apoptotic AECs with a relative paucity of caveolin-1 and increased mdm2 in fLfs. Therefore, targeting caveolin-1 using a caveolin 1 scaffolding domain peptide, CSP7, represents a new and promising approach for patients with IPF, perhaps other forms of progressive ILD or even other forms of organ injury characterized by fibrotic repair. The mechanisms of action differ in the injured AECs and in fLfs, in which differential signaling enables the preservation of AEC viability with concurrent limitation of fLf expansion and collagen secretion. The findings in three models of PF indicate that lung scarring can be nearly abrogated by airway delivery of the peptide. Phase 1 clinical trial testing of this approach in healthy volunteers has been successfully completed; Phase 1b in IPF patients is soon to be initiated and, if successful, will be followed by phase 2 testing in short order. Apart from the treatment of IPF, this intervention may be applicable to other forms of tissue injury characterized by fibrotic repair.

## 1. The Predicate for Identification of New, Well-Tolerated and More Effective Pharmacotherapy for Fibrosing Lung Diseases

Currently, there is no treatment to effectively reverse pulmonary scarring, whether chronic, as in interstitial lung diseases (ILDs), or subacute, as may occur after acute lung injuries such as acute respiratory distress syndrome (ARDS). Idiopathic pulmonary fibrosis (IPF) is a currently refractory condition characterized by progressive lung scarification and lethality, with a five-year survival of about 20 percent [1]. Food and Drug Administration (FDA)-approved drug therapy with either nintedanib or pirfenidone has been shown to slow the rate of physiologic decline or improve progression-free survival [2,3,4]. These drugs may enable improved outcomes in patients with IPF [5], but drug side effects, which generally may be gastrointestinal or related to liver function abnormalities, can be serious enough to necessitate drug cessation [6]. These same drugs have been used in other types of ILDs with beneficial effects, as recently reported [7,8,9]. Despite the availability of these new drug options, the lack of any other approved alternative agents or curative interventions for patients with progressive ILDs provides a strong predicate for the identification of more effective interventions [10,11]. While cell-based therapies have been proposed to regenerate lung tissue in IPF [12], and a number of promising targets are being tested in clinical trials [13]. None have yet been proven to be effective enough to warrant approval in IPF, additionally buttressing the argument to consider alternative potentially salutary options.

Among candidates for more effective treatment of IPF and potentially other forms of pulmonary fibrosis, targeting caveolin-1 has emerged as a promising interventional approach. Specific targeting of the caveolin scaffolding domain (CSD), evidence of tolerability, the efficacy of this approach, and formulation designed to optimally achieve airway delivery have recently been reported by our group at the preclinical level [14]. The rationale for direct airway delivery of this intervention was to limit the potential side effects and toxicity, which may be observed with the oral agents currently approved for use in patients with IPF and related fibrosing diseases of the lungs.

Based on the compelling preclinical findings in multiple models, subsequent formal toxicology analyses, and successful completion of a phase 1 clinical trial testing of a 7-mer peptide (also called LTI-03) of CSD in healthy volunteers (sponsored by Lung Therapeutics, Inc., ClinicalTrials.gov Identifier: NCT04233814) [15], a phase 1b clinical trial testing in patients with IPF will soon begin. In this review, we describe the mechanistic rationale for this new therapeutic approach, new airway and other delivery options, and the status of clinical trial testing of the lead clinical candidate for the treatment of IPF patients, CSP7.

## 2. The Contribution of Caveolin-1 to the Pathogenesis of Pulmonary Fibrosis

The pathogenesis of ILDs, including IPF, is characterized by alveolar epithelial cell (AEC) apoptosis, proliferation and accumulation of activated myofibroblasts and fibrotic lung fibroblasts (fLfs), extracellular matrix (ECM) deposition and fibrosis, resulting in progressive clinical dyspnea and loss of lung function. The caveolins are coat proteins of caveolae, which are invaginations of the plasma membranes. The enrichment of caveolae within alveolar septae suggests their potential importance in lung injury and repair. Among the caveolins, caveolin-1 is abundantly expressed by lung epithelial and endothelial cells and lung fibroblasts [16,17,18,19,20,21,22]. Caveolin-1 traffics to and from the cell membrane with multiple organelles within the cell and also as a soluble component of the cytoplasm. The CSD resides between amino acid sequences 82–101 in the NH_2_ terminal region and plays a role in caveolin-1 dimerization as well as the regulation of several signaling intermediates, many of which are implicated in the pathogenesis of lung and other organs fibrosis [20,21,22,23,24]. For example, caveolin-1 inhibits Wnt signaling and β-catenin-mediated transcription, the activation of EGFR, MEK1, and Erk2, among other signaling intermediates. The regulation of cellular signaling through caveolin-1 is cell and signaling pathway-specific [14,25,26,27,28]. Caveolin-1 may exacerbate early-phase acute lung injuries, as its expression in the injured lungs, particularly in the injured epithelium, is damaging [14,25,26,27,28]. In subsequent organizational progression with pulmonary fibrosis, caveolin-1 appears to be protective, where it exerts pro-apoptotic and anti-proliferative effects on mesenchymal cells, including lung fibroblasts and myofibroblasts [14,16,19,29,30]. Caveolin-1 expression is reduced in the lungs in IPF (Figure 1). At the same time, its’ expression in selected cell types may vary depending on the duration and severity of fibrosing injury [16,19,24,25,26,30,31,32,33,34,35]. However, caveolin-1 appears to be relatively greater in the lung epithelium, while its expression is relatively preserved in the lung endothelium and decreased in the lung interstitium in IPF or bleomycin-induced lung injury [16,17,18,36]. Caveolin-1 is increased in AECs harvested after lung injury in vitro and in vivo [24,25]. In the lung epithelium after injury, the increment in caveolin-1 appears to regulate not only AEC apoptosis but growth arrest and senescence. These effects potentiate epithelial injury, which prevents proper epithelial regeneration and stimulates fibrotic repair in fibrosing lung injuries, especially in chronic, repetitive injury as occurs in IPF. Overall, Caveolin-1 importantly interrupts TGF-β signaling and exerts potent anti-fibrotic effects through internalization and degradation of the TGF-β receptor.

An increase in caveolin-1 bioavailability, including that attributable to CSD peptides (CSPs), inhibits pro-fibrogenic signaling. Founded on these important observations, CSPs have been used to reverse cellular defects associated with fibrosing lung injury and prevent or resolve pulmonary fibrosis in animal models [14,16,21,22,23,24,25]. Along these lines, we and others have shown that CSP can block bleomycin-induced lung inflammation, injury, and fibrosis [14,16,22,37]. Systemic delivery of the 20-mer form of CSP with or without an internalization sequence mitigates bleomycin-induced lung injury and prevents the development of PF. The process involves decreased collagen deposition and reduced pro-fibrogenic signaling molecule expression, including phosphorylation of ERK, Akt, and JNK. Decreased pro-fibrogenic changes such as reduced lung expression of collagen1α1. αSMA, fibronectin, and tenascin-C to near baseline control levels also occur associated with decreased AEC senescence and apoptosis. p53 mediates differential effects on AECs and fLfs. AECs with elevated p53 typically undergo senescence and apoptosis [14,23]. By contrast, basal expression of p53 is markedly reduced in proliferating fLfs from the lungs of patients with PF, including IPF, and mice with existing PF [14,30]. CSP concurrently down-regulates p53 in injured AECs and preserves AEC viability by inhibiting senescence and apoptosis. CSP also blocks the pro-fibrogenic properties of fLfs, which proliferate and expand in IPF lungs. These fLfs demonstrate strikingly low basal levels of caveolin-1 [19,30] and p53 [30], while E3 ubiquitin-protein ligase, mouse double minute 2 homolog (mdm2) levels and mdm2-mediated degradation of p53 are markedly increased [30]. The disparities in the basal levels of p53 in injured AECs and fLfs appear to be due to increased expression of caveolin-1 in AECs undergoing senescence and apoptosis [23,24] with a relative paucity of Cav1 and increased mdm2 in fLfs [30]. It was found that CSP down-regulates p53 by inhibiting the interaction of caveolin-1 with the catalytic domain of protein phosphatase 2A (PP2A_C_) and histone deacetylase, Sirt1. This leads to increased degradation of p53 by mdm2 due to suppression of serine phosphorylation and acetylation of p53 in injured AECs. This CSP effect increases the viability of AECs. Caveolin-1 and CSP also bind to mdm2 and Sirt1, which are overexpressed in fLfs [30]. Thus, CSP inhibits mdm2-mediated degradation of p53, leading to restoration of basal p53 level in fLfs. This restrains excess proliferation and production of ECM that initiates and sustains existing PF. This shows that a treatment strategy designed to enhance the bioavailability of caveolin-1 using a CSP is salutary in these models, preserves AEC viability, and restores normal lung architecture in this setting. Further, caveolin-1 is decreased in leukocytes in bleomycin-treated mice, and CSP inhibits leukocyte recruitment into the lungs [16,36,37]. This observation is also relevant to human disease in that leukocytes from patients with scleroderma contain less caveolin-1 and more activated ERK, JNK, and p38 than normal leukocytes, while CSP reverses increased ERK, JNK, and p38 activation [37]. Overexpression of CXCR4 and MMP-9 was also decreased by CSP in leukocytes of scleroderma patients [37]. These studies highlight the central role of caveolin-1 in the pathogenesis of pulmonary fibrosis and support its potential for therapeutic targeting.

## 3. Effects of a CSD 7-mer Peptide (CSP7) on AEC Viability

In a series of prior reports [23,38,39,40,41], we showed that AEC viability is regulated by components of the urokinase-type plasminogen activator (uPA) fibrinolytic system. This process occurs in a coordinated manner in which the upregulation of uPA and its receptor, uPAR, favors AEC survival. Conversely, upregulation of plasminogen activator inhibitor-1 (PAI-1) induces senescence and apoptosis [23,38,39,40]. The changes which occur in uPA, uPAR, and PAI-1 to influence AEC viability during lung injury are, in turn, regulated solely at the posttranscriptional level. The coordinate changes of suppressed AEC uPA and uPAR with temporally and inversely increased expression of AEC PAI-1 depend upon the induction of p53 that occurs in injured AECs via caveolin-1-induced signaling intermediates. These responses demonstrate extensive cross-talk between the uPA-fibrinolytic system and caveolin-1-mediated induction of p53, in which p53 specifically binds to 3′-untranslated region sequences of uPA, uPAR, and PAI-1 mRNAs [23,38,39,40]. In bleomycin-induced pulmonary fibrosis, we found that caveolin-1 in AECs was induced with decreased AEC uPA and uPAR that occurred with concurrently increased p53 and PAI-1 [41,42]. In addition, ILDs, including IPF and emphysema, share common risk factors such as aging, premature aging due to chronic tobacco smoke, and other particulates exposure or genetic factors. The pathogenesis of ILDs, including IPF and emphysema, has been directly linked to a loss of AEC renewal capacity due to reduced viability, which limits alveolar epithelial regeneration due to stem cell exhaustion. These cells show elevated expression of caveolin-1, p53, and PAI-1, often associated with increased telomere dysfunction and endoplasmic reticulum stress. Importantly, CSP or CSP7 lacking the internalization sequence protects the lung epithelium against caveolin-1 signaling that potentiates AEC senescence and apoptosis, induction of p53, and development of pulmonary fibrosis. These findings support the use of CSP7 to inhibit AEC lung injury and prevent fibrotic remodeling (Figure 1).

## 4. Effects of CSP7 on Fibroblast Expansion and Mesenchymal Transition

Caveolin-1 deficiency promotes activation and mesenchymal transition of lung fibroblasts and increased matrix deposition by activated lung fibroblasts. Therefore, therapeutic strategies to target caveolin-1 should acknowledge cell-specific effects, including those on the lung epithelium (discussed in the previous section) and lung fibroblasts. Along these lines, we examined the responses of myofibroblasts and fLfs from patients with IPF and from mice with bleomycin-induced established pulmonary fibrosis. As illustrated in Figure 1**,** we found that p53 and microRNA-34a are relatively low in human lung fibroblasts from uninjured lungs and that levels are even reduced in fLfs from the lungs of patients with IPF or with bleomycin-induced pulmonary fibrosis due to loss of baseline caveolin-1-mediated feedback induction of expression [30,43,44]. The effects of CSP or its deletion fragment CSP7 on fLf are opposite those on AECs and involve the reversal of mdm2 catabolism of p53 [30,43,44]. p53 rises in human fLfs treated with CSP or CSP7 in vitro and in murine fLfs treated with CSP or CSP7 after induction of bleomycin-induced pulmonary fibrosis. The changes in p53 are associated with coordinate changes, in that the survival signals uPA and uPAR are suppressed as p53 levels rise in CSP- or CSP7-treated fLfs and PAI-1 levels rise, with blockade of fLf proliferation, migration, and invasion, phenotypic mesenchymal transition and expression of collagen, TGF-β and CTGF in the lungs of bleomycin-injured mice [14,24]. These observations indicate that CSP or CSP7 not only prevents apoptosis and senescence of the lung epithelium that potentiates lung fibrosis but blocks fibroblast expansion as well.

## 5. Dry Powder Inhalational Versus Nebulization or Parenteral Formulation of CSP7

CSP or CSP7 was well tolerated in the bleomycin- and TGF-β models of pulmonary fibrosis, justifying further development [14]. We found that CSP7 prevented AEC apoptosis induced by reversing the same changes in the fibrinolytic system as mentioned in the previous section [14,42]. This evidence also supports the efficacy of CSP7 in animals with bleomycin-induced pulmonary fibrosis, which is characterized by prominent AEC apoptosis. Since CSP7 targets the p53 pathway, we addressed the mutagenicity and carcinogenicity or transformational potential of this peptide using the in vitro Ames test, a bacterial-short term test often employed to detect the mutagenicity solubilized drug and the mouse micronucleus assay in vivo, respectively [14]. These studies proved that CSP7 is not mutagenic nor carcinogenic. In bleomycin-induced pulmonary fibrosis, the divergent responses of lung AECs and fLfs are seen with treatment using CSP7. Our recent publications further show that CSP or CSP7 reduces lung expression of TGF-β and CTGF to baseline levels after bleomycin-induced pulmonary fibrosis while preserving the viability of the AECs and preventing fLf expansion [14,16,21]. These changes maintain normal lung architecture with greater protection correlating with relatively more extended treatment in mice started on CSP7 at 14 days after induction of bleomycin-induced lung injury [14,30]. We recently found that CSP7 likewise confers protection in the TGF-β-induced pulmonary fibrosis model [14,44]. The protective effects of the peptide on AEC senescence/apoptosis and against fLf expansion and pulmonary fibrosis were comparable in both models. Thus, CSP7 acts on key derangements contributing to the pathogenesis of IPF as it promotes AEC viability and reduces activation and proliferation of fLfs as well as TGF-β in the injured lung.

We developed a nebulized formulation of CSP7 using a vibrating mesh nebulizer to allow direct delivery to the lung [14]. We found that CSP7 with or without lactose monohydrates or mannitol used as a drug stabilizer in nebulized formulations reduced expression of pro-fibrogenic markers such as collagen1α1, fibronectin, alpha-smooth muscle actin and tenascin-C in fLfs from IPF lungs. These responses are indicative of anti-fibrotic activity. Mice with bleomycin-induced pulmonary fibrosis were treated with nebulized CSP7 for 7 days, starting day 14 through day 20 at a dose 30 times less than the IP dose using a nose-only inhalation exposure chamber reduced existing pulmonary fibrosis, suggesting that CSP7 formulated in lactose or mannitol retains anti-fibrotic activity [14].

To avoid solubility, stability, and degradation issues associated with the liquid formulation and deliver deep lung, CSP7 was micronized by air-jet milling for dry powder inhalation [14]. We found that the dry powder inhalation of air-jet milled CSP7 from day 14–20 post-BLM exposure improved overall survival and resolved existing pulmonary fibrosis. These results show that airway delivery of CSP7 by liquid nebulization or micronized dry powder inhalation was as efficacious as IP injection in reducing bleomycin-induced existing pulmonary fibrosis at a dose approximately 30 times less than that used for IP injection. Less invasive local delivery of CSP7 appears to be more advantageous with mitigation of systemic effects in patients suffering from chronic lung diseases such as IPF.

Injured AECs isolated from fibrotic lungs, including those from patients with IPF, demonstrate a marked increase in the expression of p53 due to the induction of caveolin-1 [41,42]. However, fLfs which proliferate in IPF lungs, alternatively showed strikingly low baseline levels of p53 and caveolin-1, while mdm2 expression and mdm2-mediated degradation of p53 protein are markedly increased in fLfs. These disparities in the expression of p53 between injured AECs and fLfs appear to be due to differences in the basal expression of caveolin-1 and mdm2 in fibrotic lungs. Several of our publications demonstrate that p53 is a therapeutic target for pulmonary fibrosis through which AEC senescence and apoptosis and proliferation of fLfs can both be inhibited [14,21,30]. Further, CSP7 down-regulates p53 by inhibiting the interaction of caveolin-1 with the catalytic subunit of protein phosphatase 2A in injured AECs. This effect protects against the development of pulmonary fibrosis. Interestingly, due to the paucity of caveolin-1 and overexpression of mdm2 in fLfs, CSP7 directly binds to mdm2 and inhibits mdm2-mediated degradation of p53 leading to restoration of baseline p53 in fLfs. This inhibits excess proliferation and production of ECM proteins that cause fibrosis. We found that CSP7 in fibrotic lungs concurrently prevents the development of pulmonary fibrosis by inhibiting increased expression of p53 and apoptosis in injured AECs and by inhibiting proliferation by restoring baseline p53 expression in fLfs [30,42]. In addition, CSP7 suppresses otherwise increased expression of profibrotic cytokines such as TGF-β and CTGF, which further facilitates the progression of lung fibrosis through induction of p53 and apoptosis in AECs, and activation and production of ECM by fLfs [14,30,41,44]. All these processes appear to contribute to the ability of CSP7 to resolve established pulmonary fibrosis.

CSP7 is, therefore, efficacious in multiple mouse models of lung fibrosis when delivered intraperitoneally, nebulized, or as a dry powder during the fibrotic phase of lung injury [14]. CSP7 treatment significantly reduces ECM proteins such as collagen in mouse lung homogenate and reduces fibrosis as measured by micro-CT scan and trichrome staining by targeting the p53 master switch at the level of upstream signaling to increase the viability of injured AECs, prevents expansion of fLfs and reverses as well as prevents pulmonary fibrosis.

## 6. CSP7 Inhibits p53 Expression and Improves Viability in AECs, and Mitigates Established Pulmonary Fibrosis in Human End-Stage IPF Lung Explants Treated Ex Vivo

Treatment of IPF lung tissues with CSP7 reduced pro-fibrogenic markers with minimal, if any, effects on normal lung tissues [14]. Isolated AECs from IPF lung tissues showed a marked increase in p53, beta-galactosidase, and activated caspase-3, suggesting reduced viability due to the induction of p53 [14]. However, AECs from IPF lung tissues treated with CSP7 ex vivo showed marked suppression of p53 with improvement in viability. Importantly, a dose-dependent increase in epithelial cell survival was detected in precision-cut lung slices biopsied from both non-specific interstitial pneumonia (NSIP) and end-stage IPF patients treated with CSP7 at physiologically relevant doses. AEC survival was measured via lysotracker staining [45]. Furthermore, AEC markers ATP-binding cassette (ABC) subfamily A member 3 (ABCA3) and SP-C were increased. Interestingly, treatment of control lung tissues with CSP7 failed to alter low baseline p53 expression or apoptosis, suggesting selective targeting of injured AECs by CSP7. Similarly, in vitro treatment of AECs isolated from IPF lungs with CSP7 markedly reduced p53 expression and activation of caspase-3, while uninjured AECs isolated from non-IPF lung tissue failed to respond to CSP7, suggesting that CSP7 specifically targets injured AECs with dysregulated p53 expression. Consistent with robust anti-fibrotic responses of CSP7 in fLfs isolated from IPF lungs treated in vitro, analysis of IPF lung tissues treated with CSP7 ex vivo showed a marked reduction in pro-fibrogenic marker protein while normal lung tissues from control subjects failed to respond CSP7 treatment.

Our work using AECs and fLfs isolated from IPF lungs treated with CSP7 in vitro, intact tissues from patients with IPF, and mice with established pulmonary fibrosis treated with CSP7 ex vivo and from preclinical mouse models with both early injury and late established pulmonary fibrosis demonstrated consistent protection against progression of PF [14,24,30,41]. Based on this body of work, we strongly believe that a CSP7-based therapeutic that concurrently alters p53 expression in injured AECs and in fLfs will improve the outcomes of IPF or other ILDs in an effective and well-tolerated way. Anti-fibrotic effects of CSP7 on both AECs, fLfs, and fibrotic lungs, including IPF lungs, render it a strong candidate for IPF therapy. Given the paucity of any effective treatment to cure fibrotic lung disease, there is a significant market opportunity and value proposition for developing CSP7 in this orphan indication.

## 7. Alternative Targetable Pathways May Substantively Contribute to the Pathogenesis of Pulmonary Fibrosis

While CSP7 has now emerged as a promising candidate for the treatment of pulmonary fibrosis in humans, many other targets for the treatment of pulmonary fibrosis remain under active investigation or have entered clinical trial testing [13]. CSP7 inhibits TGF-β, CTGF, and lung inflammation, as addressed in the sections above. It is an especially attractive intervention because of the differential effects it exerts to suppress myofibroblast proliferation and alveolar apoptosis [14,46]. However, it is possible that CSP7 may disproportionately exert effects on TGF-β, it’s signaling, or other intermediates that may likewise account for the known salutary effects on CSP7 in murine models of pulmonary fibrosis [14]. Because of its ability to suppress fibrosis in a number of organs, targeting of TGF-β1 pathway remains under investigation. Receptor-TGF-β1 targeting is problematic because of the potential for broad adverse effects, but the targeting of downstream intermediates may prove to be more selective, well-tolerated, and effective [47].

The recent literature is replete with studies in which a range of therapeutic targets have emerged as candidates for the treatment of pulmonary fibrosis, and IPF in particular. Examples of some very novel and interesting approaches include the identification of novel targeted strategies targeting collagen itself to inhibit its deposition in the neo-matrices found in the organizing lung [48]. Another potential approach involves the targeting of histone deacetylases, which may promote cellular proliferation and resistance to apoptosis in neoplasia. Aberrant overexpression has been found in the lungs of patients with IPF, raising the possibility that histone deacetylases may represent novel targets to interdict fibrogenesis [49]. Another new and interesting target is VISTA, the myeloid immune regulator PD-1H, which is overexpressed in monocytes infiltrating the lungs of subjects with IPF [50]. VISTA deficiency promoted pulmonary fibrosis in bleomycin-treated mice, while a VISTA agonist antibody ameliorated this response, demonstrating interventional feasibility and supporting the predicate for further investigation. Another interesting and promising approach involves the use of next-generation miR-29 mimic with improved bio-availability, which reduced profibrotic responses in several model systems, including human precision-cut lung slices and bleomycin-induced pulmonary fibrosis, was well-tolerated in toxicology studies [51]. HER2 (receptor tyrosine-protein kinase erbB-2) signaling has also recently been identified as a potential target for intervention in IPF, based upon the identification of HER2 as a key element of the activation signature of invasive lung fibroblasts from IPF subjects and induction of pulmonary fibrosis with overexpression of HER2 in normal lung fibroblasts [52]. These selected studies demonstrate that a number of candidate therapeutic targets are being identified and are at differing stages of the translational trajectory. Like CSP7, all must undergo rigorous clinical trial testing in patients with IPF or other fibrosing lung conditions to conclusively demonstrate their efficacy and safety. Such strategically targeted CSP7 included, may eventually emerge as candidate interventions for fibrosis in other organs as well. While it is conceivable that several targeted strategies will eventually demonstrate efficacy and safety in patients with IPF or related conditions, the identification of optimal interventional agents and perhaps combinatorial or personalized targeted therapy remains a top priority within the field.

## 8. Translation of CSP7 to Phase 1 Clinical Trial Testing for Pulmonary Fibrosis

The initial work that enabled preclinical testing of CSP7 was supported by the National Heart, Lung, and Blood Institute of the National Institutes of Health. A US patent for the use of CSP7 for the treatment of pulmonary fibrosis was filed in March 2009 and allowed in 2013, with subsequent related patents allowed in 2017 and 2021. The patents were licensed by the University of Texas Board of Regents to Lung Therapeutics incorporated (LTI), which subsequently began commercialization of the peptide. LTI’s initial commercialization steps included bulk manufacturing, formulation for airway delivery, and formal toxicology studies in anticipation of subsequent clinical trial testing. In January 2020, LTI sponsored a phase 1 clinical trial of CSP7 delivered to the airways by dry powder inhalation in healthy adult individuals (ClinicalTrials.gov Identifier: NCT04233814). Solicitation of NIH funding was considered for this trial, but successful fundraising by the company enabled corporate sponsorship with an arguably more rapid trajectory toward initiation. Initial testing included a single ascending dose (SAD) format to be followed by additional cohorts in which multiple ascending doses (MAD) will be tested. The study was designed to test the safety and tolerability of the inhaled agent and pharmacokinetics. Pulmonary function is being used to screen for the development of bronchospasm. Chronic obstructive pulmonary disease and asthma were exclusion criteria. A micronized lactose powder placebo serves as the control comparator to CSP7 administered via the identical airway delivery system. The phase 1 study has been completed, and aerosolized CSP7 dry powder was preliminarily found to be well-tolerated with no significant impact on respiratory function [15]. If the dose escalations demonstrate that dry powder delivery of CSP7 is well-tolerated and safe, then a phase 2 trial will next be conducted, to be initiated within 12 months of the conclusion of the phase 1 trial. LTI has assembled a team of internationally recognized experts to advise the company about the most appropriate design of the phase 2 trial in which the intervention will be tested for efficacy and safety in a cohort of patients with IPF.

## 9. Potential Applications of CSP7 for Organization and Fibrosis in Other Organs

The beneficial effects of anti-fibrotic therapy with pirfenidone in patients with IPF have led to trials in other forms of tissue injury followed by fibrotic repair. For example, pirfenidone has been shown to alleviate renal tubulointerstitial fibrosis, and the effect involves the inhibition of miR-21 [53]. Pirfenidone has also been proposed as a candidate treatment to prevent neomatrix formation and cardiac fibrosis [54]. Interestingly, a small study recently suggested that the addition of pirfenidone to corticosteroid therapy attenuated lung restriction and improved CT chest imaging in patients with COVID-19 [55]. Pirfenidone has also been shown to prevent fibrosis in preclinical forms of liver disease [56,57]. Nintedanib has likewise been shown to exert anti-fibrotic effects in models of skin, liver, and renal fibrosis as recently reviewed [58].

These reports suggest that anti-fibrotic therapy may be applicable in a variety of forms of tissue injury characterized by fibrotic repair. Whether the currently approved agents can be more broadly applied in clinical practice remains to be seen. The ability of CSP7 to inhibit non-pulmonary forms of tissue fibrosis awaits further preclinical assessment. However, salutary effects of CSP or related peptides in other organ inflammation and fibrosis and in tumor progression through either inhibition of tumor microvascular permeability or targeting tumor-associated caveolin-1-deficient stromal fibroblasts, which feed cancer cells, have been reported in a variety of models [59,60,61,62]. Similarly, several recent independent studies [63,64,65,66,67,68,69] demonstrate that caveolin-1-dependent pathways have beneficial effects on Crohn’s disease, colitis, and cardiac, renal, and liver fibrosis by inhibiting angiogenesis, barrier dysfunction, autophagy, inflammation, endoplasmic reticulum (ER) stress and TGF-β1/Smad signaling, suggesting the broader therapeutic utility of CSP7 in organs with different cellular composition. In addition, recent literature [21,36,70,71,72,73] suggest that caveolin-1 or related peptide has beneficial effects on various other lung diseases such as mucus hypersecretion and emphysema associated with COPD, asthma and airway remodeling, pulmonary hypertension, sleep apnea, bronchopulmonary dysplasia, and hyperoxic lung damage. The ability of other small molecular inhibitors to prevent lung, renal, and hepatic diseases suggests the possibility that others may be broadly applied to possible clinical advantage [74,75].

## Figures and Tables

**Figure 1 cells-12-00554-f001:**
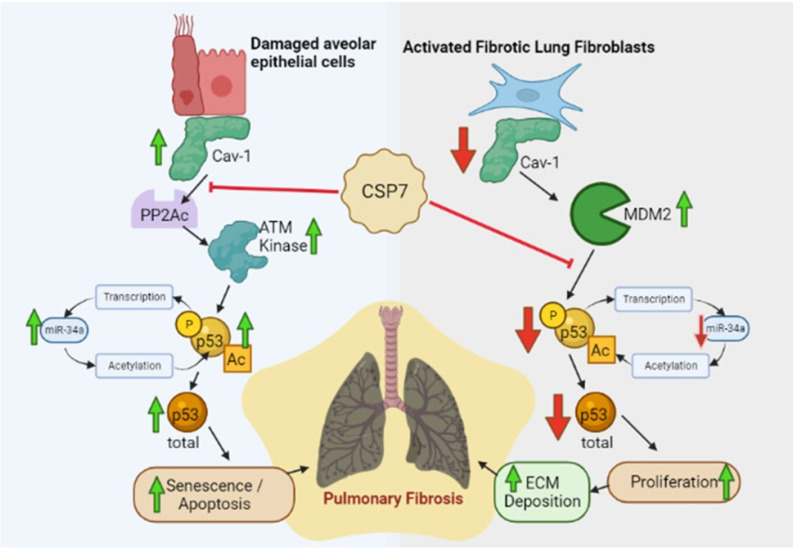
Differential effects of CSP7 on injured AECs and activated fLfs. Competitive inhibition of Cav1 signaling in injured AECs and concurrent complementation of Cav1 signaling in fLfs by CSP7 alleviate disparities in basal p53 levels and resolve existing PF.

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
