# Peer review of "Caveolin-1-Related Intervention for Fibrotic Lung Diseases"

_cells, 2023, doi:10.3390/cells12040554_

Round 1

Reviewer 1 Report

This is a very focused review that mainly summarizes the authors’ work on the role of caveolin-1 targeting CSP7 in lung fibrosis. The wealth of evidence from the authors’ group over the years suggests CSP7 is a promising drug candidate.

I have only several minor suggestions:

Line 108: references needed.

Line 134 and 153, the discussion should be expanded to include how CSP7 has differential regulation on p53/mdm2 in AEC and lung fibroblasts.

Line 354: rationale for CSP7 to treat fibrotic diseases in other organs needs to improve. Citing pirfenidone may not justify the use of CSP7 because the uniquely differential expression of the CSP7 target, caveolin-1, in AEC and lung fibroblasts may not exist in other organs. The regimen of CSP7 to treat fibrotic diseases in other organs needs some more discussion because it is inhaled in the clinical trial for lung fibrosis.

May discuss a little more on the potential therapeutic effect of this peptide on other pulmonary diseases in which caveolin-1 may also have a role.    

Author Response

Reviewer 1 Comments:

  • English language and style are fine/minor spell check required.

Response: Agreed.

  • This is a very focused review that mainly summerises the authors work on the role of caveolin-1 targeting CSP7 in lung fibrosis. The wealth of evidence from the authors’ group over the years suggests CSP7 is a promising drug candidate.

Response: We appreciate the Reviewer’s favorable assessment our work.

  • Line 8: references needed.

Response: Agreed and made suggested changes to the text.

  • Line 134 and 153, the discussion should be expanded to include how CSP7 has differential regulation on p53/mdm2 in AEC and lung fibroblasts.

Response: Agreed and added details (please see lines 124-142) to the revised text.

  • Line 354: Line 354: rationale for CSP7 to treat fibrotic diseases in other organs needs to improve. Citing pirfenidone may not justify the use of CSP7 because the uniquely differential expression of the CSP7 target, caveolin-1, in AEC and lung fibroblasts may not exist in other organs. The regimen of CSP7 to treat fibrotic diseases in other organs needs some more discussion because it is inhaled in the clinical trial for lung fibrosis.

Response: We appreciate Reviewer’s suggestion. Accordingly, we made the suggested changes with additional information (Please see lines 384-395).

  • May discuss a little more on the potential therapeutic effect of this peptide on other pulmonary diseases in which caveolin-1 may also have a role.  

Response: Agreed. Accordingly, we revised the text (please see lines 392-395) with additional information describing the potential therapeutic benefits of CSP7 on other lung diseases as suggested.

Reviewer 2 Report

The manuscript by Shetty and Idell explores the role of caveolin-1 in fibrotic lung disorders.  In general, the manuscript is well written and proceeds logically.  The authors are experts in this area and have contributed greatly to our understanding of this field.  Thus, the comments made below are meant to strengthen it:

1)  The divergent roles of caveolin-1 in AECs and fibroblasts is a key aspect of the presentation.  Perhaps the distinction should be highlighted earlier to prepare the readers for a more detailed description provided later.

2) Age, tobacco exposure, and genetics are considered important host factors involved in the pathogenesis of pulmonary fibrosis.  The authors are encouraged to discuss how the processes presented here may relate to processes related to aging, tobacco, and genetic factors in lung fibrogenesis.

Others:

a) There are a few typos here and there that need revision.

b) The abbreviation mdm2 should be spelled completely in the abstract.

c) Page 5, line 181:  Please describe what is Ames

Author Response

Reviewer 2 Comments:

  • English language and style are fine/minor spell check required.

Response: Agreed. Please see response to Reviewer 1.

  • The manuscript by Shetty and Idell explores the role of caveolin-1 in fibrotic lung disorders.  In general, the manuscript is well written and proceeds logically.  The authors are experts in this area and have contributed greatly to our understanding of this field.  Thus, the comments made below are meant to strengthen it:

Response: We are gratified by the Reviewer’s assessment of our contribution to the field and appreciate helpful comments to strengthen the presentation.

  • 1) The divergent roles of caveolin-1 in AECs and fibroblasts is a key aspect of the presentation.  Perhaps the distinction should be highlighted earlier to prepare the readers for a more detailed description provided later.

Response: Agreed. Please see highlighted lines 124-142 in the revised text.

  • 2) Age, tobacco exposure, and genetics are considered important host factors involved in the pathogenesis of pulmonary fibrosis.  The authors are encouraged to discuss how the processes presented here may relate to processes related to aging, tobacco, and genetic factors in lung fibrogenesis.

Response: Agreed. Accordingly, we revised the text (please see lines 166-173) with additional discussion of host factors such as age, exposure to tobacco smoke and genetics in lung fibrogenesis diseases as suggested.

Others:

  • a) There are a few typos here and there that need revision.

Response: Agreed.

  • b) The abbreviation mdm2 should be spelled completely in the abstract.

Response: Agreed. Please see revised lines 25 and 130-131.

  • c) Page 5, line 181:  Please describe what is Ames

Response: Agreed. We have now described Ames test in revised lines 204-206.